# Tribo-Mechanical Investigation of Glass Fiber Reinforced Polymer Composites under Dry Conditions

**DOI:** 10.3390/polym15122733

**Published:** 2023-06-19

**Authors:** Corina Birleanu, Marius Pustan, Mircea Cioaza, Paul Bere, Glad Contiu, Mircea Cristian Dudescu, Daniel Filip

**Affiliations:** 1Micro Nano Systems Laboratory, Technical University from Cluj-Napoca, Blv. Muncii Nr. 103–105, 400641 Cluj-Napoca, Romania; corina.barleanu@omt.utcluj.ro; 2Department of Manufacturing Engineering, Technical University from Cluj-Napoca, 400641 Cluj-Napoca, Romania; paul.bere@tcm.utcluj.ro (P.B.); glad.contiu@tcm.utcluj.ro (G.C.); 3Department of Mechanical Engineering, Technical University from Cluj-Napoca, 400641 Cluj-Napoca, Romania; mircea.dudescu@rezi.utcluj.ro; 4Department of Management and Economic Engineering, Technical University from Cluj-Napoca, 400641 Cluj-Napoca, Romania; daniel.filip@mis.utcluj.ro

**Keywords:** GFRP/epoxy, vacuum bag technology, tribology properties, friction coefficient COF, dry abrasion wear, wear rate, sliding velocity

## Abstract

Tribo-mechanical experiments were performed on Glass Fiber Reinforced Polymer (GRFP) composites against different engineering materials, and the tribological behavior of these materials under dry conditions was investigated. The novelty of this study consists of the investigation of the tribomechanical properties of a customized GFRP/epoxy composite, different from those identified in the literature. The investigated material in the work is composed of 270 g/m^2^ fiberglass twill fabric/epoxy matrix. It was manufactured by the vacuum bag method and autoclave curing procedure. The goal was to define the tribo-mechanical characteristics of a 68.5% weight fraction ratio (wf) of GFRP composites in relation to the different categories of plastic materials, alloyed steel, and technical ceramics. The properties of the material, including ultimate tensile strength, Young’s modulus of elasticity, elastic strain, and impact strength of the GFPR, were determined through standard tests. The friction coefficients were obtained using a modified pin-on-disc tribometer using sliding speeds ranging from 0.1 to 0.36 m s^−1^, load 20 N, and different counter face balls from Polytetrafluoroethylene (PTFE), Polyamide (Torlon), 52,100 Chrome Alloy Steel, 440 Stainless Steel, and Ceramic Al_2_O_3_, with 12.7 mm in diameter, in dry conditions. These are commonly used as ball and roller bearings in industry and for a variety of automotive applications. To evaluate the wear mechanisms, the worm surfaces were examined and investigated by a Nano Focus—Optical 3D Microscopy, which uses cutting-edge μsurf technology to provide highly accurate 3D measurements of surfaces. The obtained results constitute an important database for the tribo-mechanical behavior of this engineering GFRP composite material.

## 1. Introduction

In an increasingly technological world, composite materials play a vital role. Due to their advantageous properties such as high mechanical strength, wear resistance, anti-corrosive properties as well as low weight [1,2,3], glass fiber reinforced polymer (GFRP) composites are widely used in the military, aerospace, automotive industry, at the vanes of windmills and pumps, etc. Each type of GFRP structure has unique properties and is used for various applications. In the specialized literature, the tribological, mechanical, thermal, water absorption, etc., properties of different polymer composites reinforced with Glass Fiber (GF) have been reported.

Because of their favorable tribological characteristics, polymer composites reinforced with glass fiber are frequently employed in engineering applications. Some of these composites’ essential tribological characteristics include:

(a)GFs are incredibly tough and give the polymer matrix great wear resistance as a result. In situations where there is a lot of sliding or rubbing between two surfaces, these composites are excellent choices.(b)Low friction coefficient: Because GFRP composites have a low friction coefficient, they operate more efficiently and produce less heat, which results in less wear and longer service life for the component.(c)High load-bearing capacity: Glass fibers’ high tensile strength enhances the polymer matrix’s load-bearing capacity and stiffness, enabling it to tolerate greater loads and stresses without permanently deforming.(d)Excellent dimensional stability: Glass fibers’ low coefficient of thermal expansion gives the composite material its good dimensional stability. By doing this, it ensured that the component would stay its original size and shape even in extreme heat and other environmental circumstances.

Ya-Jung Lee et al. [4,5,6,7,8] used polyester resin with glass-fiber-reinforced fillers to analyze the mechanical properties, including tensile strength, flexural strength, and Young’s modulus for single and multiple fibers. They have created a numerical model using trial data, which might be used in the future to access the mechanical properties of laminates delivered using various fiber kinds without extra tests. They also concluded that when flexural loading is applied to GFRP laminates, the stiffness after initial failure does not immediately decrease but instead degrades gradually. The vacuum infusion processed GFRP samples showed better mechanical properties than the hand layup technique because the hand layup technique increased porosity. Sang-Young Kim et al. [9] used vinyl ester epoxy resin and glass fiber reinforced fillers to analyze mechanical properties such as tensile strength, compressive strength, and in-plane shear properties.

To decrease weight, improve corrosion resistance, and boost the strength-to-weight ratio, Selvaraju et al. [10,11,12,13,14] examined the use of Aramid Fiber Reinforced Polymer (AFRP) and GFRP in marine applications like naval patrol boats, submarine diesel storage tanks, lube tanks, utility tanks, low-pressure pipes, cable ladders, and trays. Chalmers [15] investigated several combinations of reinforcements, including E-glass, carbon, and aramid fiber, with various types of resins, including polyester resins, vinyl ester resins, and epoxy resin, to minimize stiffness, lower maintenance requirements, and lower production costs. The various composite materials used in marine construction, such as GFRP, CFRP, AFRP, and hybrid composite materials (Fiber Reinforced Polymer-FRP with two or more different types of fibers), have been studied by Isao Kimpara [5,12,13,14,16], including their current and future applications. According to the study’s findings, GFRP has primarily been used in the marine environment, where weight reduction, boat speed, and mechanical and dynamic strength against creep, impact, and fatigue are all improved.

Friction and wear of polyimides reinforced with carbon, glass, and aramid fibers were examined by Gai Zhao et al. [17,18]. According to the research, reinforcement enhances the tribological characteristics of composites [19,20,21,22], and inorganic-fiber-reinforced composites outperformed glass-fiber-reinforcement in terms of performance because they effectively shared load between surfaces in contact. A study by Suresha et al. [19] used epoxy resin with reinforced E-Glass and epoxy composites filled with graphite. The wear qualities of epoxy with E-Glass filler were found to be reduced, and the epoxy composites with E-Glass and graphite showed the least wear when compared to neat epoxy. The wear resistance was measured for three distinct combinations.

When E-Glass and graphite-filled composites were added, the mechanical parameters, such as tensile strength, tensile modulus, and hardness, improved. Overall, the use of glass fiber as a reinforcement in polymer composites offers important benefits in terms of improved tribological performance, making them appropriate for use in a variety of applications, where high-performance materials are required.

An in-depth analysis has not yet been performed, and more research is needed to investigate and build the most complete database possible. This research will focus on the mechanical and tribological behavior of various polymer material recipes that slide against various types of materials under various normal loads and sliding speeds.

The novelty of the research presented in this study lies in investigating the tribomechanical properties of a GFRP/epoxy composite. The investigated composite material differs from the other materials presented in the literature by the type of reinforcement material, the type of matrix, the degree of reinforcement, the polymerization process as well as the arrangement of the layers. Therefore, in this study, the main objective is to analyze the sliding friction properties of own GFRP-made material couplings with metal, technical ceramic, Polytetrafluoroethylene (PTFE), and Torlon under low load and low-speed conditions in dry lubrication conditions with respect to the sliding direction.

## 2. Glass Fiber Reinforced Polymer Composites (GFRP)—Manufacturing and Properties

### 2.1. Fabrication Procedure of GFRP Sample

For the experiment, a composite material designed from a thermoset polymer matrix reinforced with a 270 g/m^2^ GF twill fabric was used.

The polymer matrix was epoxy resin type EPIKOTE™ Resin MGS LR 135 (HEXION GmbH, Duisburg, Germany) commonly used for glass, carbon, or aramid fiber processing, and EPIKURE™ Curing Agent MGS LH136 hardener with a mix ratio of 100: 35 ± 2 g. To manufacture the GFRP plate testing, the vacuum bag technology and autoclave curing procedure presented in [23,24,25,26,27] were used. The GF layers were impregnated by the epoxy resin (wet technology) using the hand lay-up method. A metal mold was used.

After the simultaneous application of the resin layers and the fabric strips, the entire surface was manually rolled to eliminate the accumulation of air bubbles between the overlapping layers.

In the next step, the GFRP was covered with a perforated film and an absorbent breather fabric. The GFRP and mold were placed in a vacuum bag which was sealed by welding the edges using a hot wire installation. The bag was subjected to a vacuum pressure of −0.9 bar. After applying this pressure, the air microbubbles and the excess resin from the composite were transferred to the absorbent breather. These auxiliary materials have the role of allowing air and excess resin to be removed from the composite material (Figure 1).

Additionally, the vacuum bag (mold, composite, auxiliary materials) was introduced into the Maroso autoclave (Maroso SRL, Pianezze, Italy) (Figure 2), where it was subjected to a curing cycle. The autoclave is a pressure chamber specifically built to guarantee the optimization of the curing processes carried out through the constant control of the temperature, pressure, and vacuum parameters during the whole cycle, with absolute safety and respect for the operating conditions.

In the first step, the temperature was increased from 0 °C to 80 °C in 30 min, applying an additional pressure of 4 bars. In the second step, the temperature was increased to 120 °C over the course of 20 min and the pressure was maintained at 4 bars. In the third stage of the cycle, the temperature was maintained at 120 °C and the pressure at 4 bars for 2 h. In the last step of the cycle, the pressure was reduced to 0 bar, the vacuum plant was stopped, and the temperature was brought to 60 °C in a 30 min interval.

Following the aforementioned process, a GFRP material with dimensions of 400 mm× 295 mm × 2 mm was obtained, with a weight fraction ratio of 68.5%. This material was used to process discs with a diameter of Ø 50 mm for use in the experiment (as depicted in Figure 3).

### 2.2. Mechanical Properties of GFRP

The mechanical properties of GFRP were evaluated by tensile, impact, and hardness measurements. Standard procedures ASTM D638-14 (Standard Test Method for Tensile Properties of Plastics) and ASTM D790-03 (Standard Test Methods for Flexural Properties of Unreinforced and Reinforced Plastics) were followed to analyze the tensile and flexural properties of the GFRP materials for five specimens.

The bending samples were evaluated by a three-point flexural test using an Instron 3366–10 kN (Instron, Norwood, MA, USA) universal test machine. The load was applied at a testing speed of 2 mm/min until the specimen broke. Table 1 lists the mean values of flexural mechanical proprieties and standard deviation between evaluated specimens.

Tensile specimens were tested on an Instron 8801–100 kN (Instron, Norwood, MA, USA) servo-hydraulic testing machine. The strain data were collected based on transverse displacement. Figure 4a,b show the flexure and tensile stress–strain curves, while Table 2 presents the mean values of tensile mechanical proprieties and standard deviation between evaluated specimens.

The evaluated mechanical proprieties (strength, strains, and modulus of elasticity) reflected by their standard deviation showed good reproducibility of the mechanical tests and relevance of the results.

To assess the mechanical quality, specifically the hardness, of the glass fiber-reinforced composite disc, a Mitutoyo HR-430 Series Digital Rockwell Hardness Tester (Mitutoyo Europe GmbH) was utilized. Five specimens, each with dimensions of 30 mm × 30 mm, were cut from the composite material. The hardness testing was performed under natural atmospheric conditions in a laboratory setting. Each specimen was indented at least 3 mm from both its edge and any other indentation, and only smooth surfaces co-ered in polyester resin were indented. Five readings were taken from each specimen and recorded. The resulting average Rockwell Scale Ball 1.5875 mm hardness value was HRF 81.60 (Figure 5).

### 2.3. Ball Materials

For the wear tests, a bearing ball made of various materials was utilized as a counterpart. The standard balls used in the test were PTFE (Teflon), Torlon (polyamide), 52100 Chrome Alloy Steel, 440 Stainless Steel, and Ceramic Al_2_O_3_, all of which had a diameter of 12.7 mm (Figure 6). Of these, 52100 Chrome Alloy Steel, 440 Stainless Steel, and Ceramic Al_2_O_3_ are frequently employed in industry as ball and roller bearings and are well-regarded for their excellent surface quality, superior wear resistance, hardness, and high load capacity. Table 3 provides information on the chemical composition characteristics and mechanical properties of the balls used in this investigation.

## 3. Experimental Method and Device

The wear tests were carried out on a pin-on-disc tribometer (Figure 7), in dry conditions, in an ambient environment at 22 ± 1–2 °C, and relative humidity of 45 ± 1–2%.

An electrical motor rotated the disc 50 mm in diameter and 2 mm in thickness. The pin was replaced with a bearing ball with a diameter of 12.7 mm, adapted to the clamping system.

In each test, a new ball and disc were used, and prior to starting the test, they were cleaned and wiped dry using a technical cleaner. The test lasted for 120 min, during which the friction coefficient was continuously recorded to determine the running-in friction regime and the friction coefficient in the steady-state regime.

The wear loss of the ball or disc was determined gravimetrically through microscopic examination and surface profiling.

The sliding velocities and normal load were set to 0.10, 0.25, and 0.36 m s^−1^ and 20 N, respectively.

The temperature, wear, and friction force were continuously measured throughout the test with a tolerance of 2–3%. The discs were machined to have an outside diameter of 50 mm and a thickness of 2 mm. The roughness parameter Ra was measured based on ISO GPS standards using a Gaussian filter on a sampling length of 0.8 mm, and an evaluation length of 4 mm.

The surface roughness of the discs was measured using the ISR-C300 INSIZE Detachable Probe Rugometer, and the average roughness parameter Ra value was found to be 0.38 µm based on multiple measurements. The roughness values of the balls are presented in Table 3, which can aid in developing precise wear models and conducting predictive tribological evaluations.

A 12.7 mm ball made of a different material was mounted in an adapting support, which was held in place by an arm. A force cell and a digital indicator were used to measure the frictional force. The friction coefficient was calculated by dividing the measured friction force by the applied normal load. To ensure the reliability of the test results, the experiment was repeated three times and the average value was considered. The spread of the results was described by the error bars, which indicate the corresponding standard deviation.

After a certain number of cycles, the test was stopped, and the surface topography was analyzed to quantify the amount of wear and track the development of surface roughness. To obtain more information about the wear phenomenon and the mechanisms for material removal, the worn surfaces of the ball and the track surface of the disc were analyzed using Optical 3D microscopy (OM) after each test.

The surface modifications were investigated by a Nano Focus—Optical 3D Microscopy, which uses cutting-edge μsurf technology to provide highly accurate 3D measurements of surfaces. It is based on continuous focus variation technology with fixed focal length goals, 10x zoom was the scanning target. Radial layers of the scanned surface were extracted and used to examine the surface.

Using an analytical microbalance with a 0.1 mg precision, the weight loss of the ball and disc was used to measure wear. According to the following equation, the wear rate (K, mm^3^/Nm) observed in this investigation was calculated:(1)$$K=\frac{m}{L_{sliding}\cdot F\cdot \rho }=\frac{V}{L_{sliding}\cdot F}$$
where m is the mass loss (mg), L*_sliding_* is the sliding distance (m), F is the applied load (N), V is the loss volume (mm^3^), and ρ is the density of material samples (g/cm^3^). To reduce data scattering, three replicate sliding tests were performed in this study; the results’ average values for the friction coefficient and wear rate were used.

In cases where the difference in weight loss due to wear could not be discerned, a 3D optical profilometer was employed to determine the profile of the wear tracks (as shown in Figure 8). The wear volume of the samples was calculated by measuring the wear width and depth using the profilometer. The wear track model used to determine the wear volume is illustrated in Figure 9.

The computations were performed using empirical mathematical equations, presuming that the ideal ball geometry that forms the wear scars is true. The calculation for the track’s zone area was carried out in accordance with Figure 9. The following equation was used to compute the worn track’s overall volume:(2)$$V=\frac{\pi \cdot h^{2}}{6}\cdot \left(3\cdot b^{2}{+h}^{2}\right)\cdot L$$
where V is the total volume loss of the wear track (measured in mm^3^), L is the stroke length (measured in mm), R is the radius of the ball (6.35 mm), r is the radius track (16 mm), b is the wear track’s width (in mm), h is the triangle’s height (measured in mm), and h is the depth of the wear profile (in mm).

With a 160 × 120 IR pixel resolution, the Flir E30 infrared camera was used to measure the temperature during the tests. The temperature range of the camera is −20 to 250 °C, with a thermal sensitivity of 100 mK, or less than 0.1 °C. The testing parameters are presented in Table 4.

## 4. Experimental Results and Discussion

### 4.1. Sliding Speed Effect on Friction and Wear

Two materials were tested in the initial phase of the study: a glass fiber composite disc and a ball made of PTFE (polytetrafluoroethylene). These materials are frequently utilized in engineering applications due to their exceptional properties. PTFE, for example, is a type of fluoropolymer that is recognized for its chemical inertness, non-stick properties, and low friction coefficient. The frictional properties of PTFE are influenced by various factors, such as the surface roughness of the contact area, the contact pressure, and the sliding velocity. To determine the materials’ behavior, tests were performed at three different sliding speeds (0.1, 0.25, and 0.36 m/s) and with a load of 20 N.

While contact stress between mating parts is usually not a major issue, it can cause significant problems in some cases if it is not considered. The contact between a sphere and an elastic half-space can be described using the formulas of Hertzian theory [27,28,29].
(3)$$\sigma_{c(\max)}≃0.4\cdot {\left(\frac{E_{}^{*2}\cdot F}{R^{2}}\right)}^{\frac{1}{3}}$$
(4)$$\frac{1}{E_{}^{*}}=\frac{1}{2}\left(\frac{1-\nu_{1}^{2}}{E_{1}}+\frac{1-\nu_{2}^{2}}{E_{2}}\right)$$

E_1_, E_2_ are the elastic moduli and ν_1_, ν_2_ are the Poisson’s ratios associated with each body (different ball/GRFP disc). In this case $\sigma_{c(\max)}=\;0.7363\;\mathrm{MPa}$.

The graphs presented in Figure 10 show that the coefficient of friction (COF) increases slightly at the start of the test and then and then stabilizes over time. This behavior is common in friction measurements and is attributed to the run-in phenomenon, which is typical for composite material/PTFE coupling. During the run-in period, the surface topography and chemistry of the materials change until the friction system reaches a state of equilibrium. The final COF value at this equilibrium state is usually reported.

After testing the glass fiber composite disc and the ball made of PTFE, it was observed that the surface changes on the GFRP disc were minimal. The coefficient of friction (COF) values stabilized within a few minutes after the start of the test, and for a force of 20 N, the COF was found to stabilize after 10–15 min of testing.

A similar variation in the COF also appears due to the change in the sliding speed (Figure 10, namely, at the beginning of the test, the measured coefficient of friction (COF) is slightly lower and then increases progressively. The coefficient of friction of PTFE decreases with increasing speed. This behavior may be attributed to the formation of a self-lubricating film on the surface of PTFE when in contact with other materials. The membrane is formed by the shear-induced orientation of PTFE polymer chains, reducing adhesion between contact surfaces, and lowering the coefficient of friction.

Experimental investigations have revealed that the friction coefficient of PTFE decreases from 0.215 to 0.160 as the sliding speed increases from 0.10 to 0.36 m s^−1^. The literature suggests that the friction coefficient of PTFE decreases by approximately two orders of magnitude as the speed increases from zero to hundreds of revolutions per minute. However, above a certain speed, the friction coefficient stabilizes, indicating that a self-lubricating film has been formed and has reached a steady state.

It should be noted that the exact change in PTFE’s coefficient of friction with speed is dependent on several factors, including the surface finish of the mating surfaces, applied loads, and environmental conditions. Thus, specific tests are necessary to determine the frictional behavior of PTFE under specific conditions of interest.

Regarding the local temperature, the variance has stabilized due to the ball on disc fault after 25 to 30 min of testing in all speed scenarios (Figure 10), and the temperature does not exceed 52–53 °C. Experiments show that the temperature decreases from 52–53 °C to 36–37 °C as the speed increases from 0.10 to 0.36 m s^−1^.

The friction and wear rate of GF-reinforced composite materials can vary depending on several factors, including the sliding speed. Generally, at higher sliding speeds, there is an increase in both friction and wear rate due to increased heat generation and contact pressure.

The second couple of materials investigated are the GF composite disc and the ball made of Torlon 4200. Torlon is a high-performance thermoplastic material that is known for its excellent mechanical and thermal properties, including high strength, stiffness, and anti-wear resistance. The contact stress based on Equation (3) is $\sigma_{c(\max)}=1.33\;\mathrm{MPa}$.

The COF for force 20 N stabilized after 45–60 min of testing. The experiment indicated that the friction coefficient of Torlon decreases by 0.58 to 0.52 as the speed increases from 0.10 to 0.36 m s^−1^ (Figure 11). The COF in dry sliding conditions under low load is relatively high due to the high surface roughness and adhesion between materials. The exact value of the COF for Torlon against GFRP can vary depending on the specific test conditions and the method used to measure it. Therefore, it is important to carefully design and conduct tribological tests to obtain accurate and meaningful results.

Dry sliding generates significant amounts of heat, and it is reported that the temperature rises during the test of Torlon against GFRP especially at 0.36 m s^−1^ speed, around 110 °C (Figure 11). The temperature stabilized for all three sliding speed values after 50–60 min of testing. As a result, it is crucial to carefully plan and carry out tribological tests to control the temperature rise and prevent any thermal damage.

For the glass fiber composite disc and the ball made of 52,100 Chrome Alloy Steel, the testing results are presented below. The contact stress based in this case is $\sigma_{c(\max)}=4.438\;\mathrm{MPa}$.

The effect of sliding speed on the dry wear behavior of GFRP against 52,100 Chrome steel is shown in Figure 12. The friction coefficient of the pair of materials decreases as the sliding speed increases. The wear rate and friction also decrease with increasing velocity, attributed to the presence of wear particles on the sliding surface. The mass loss of both chrome alloy steel and GFRP increases with increasing velocity. The dominant wear mechanism for the alloy steel is delamination wear, as evident from the color variations in the 3D graph.

Due to the continuous removal of wear debris during wear, the coefficient of friction is descending with respect to speed.

The heat generated by friction can result in a plastic state at the contact surface and the pores may become filled with debris produced during the initial stages of wear. These phenomena may contribute to lower frictional forces at higher working parameters. Additionally, elevated heat generation levels can lead to increased chemical reactions between the contact surface and the environment, resulting in the formation of a hard oxide layer at the surface of the specimen. This layer can act as a barrier to further material removal, thereby increasing the frictional coefficient. The oxide layer persists up to a certain threshold speed value, after which it is detached from the surface due to increasing speed, and the process of material removal from the contact surface continues. As a result, the frictional coefficient decreases after exceeding this threshold speed value. Depending on the test conditions and time stage, the minimum and maximum observed values of the coefficient of friction for the alloy steel were 0.11 and 0.415, respectively.

The influence of sliding speed on dry wear properties of GFRP against 440 Stainless Steel is presented in Figure 13. The minimum and maximum values of the coefficient of friction for the stainless steel are observed to be 0.12 and 0.485, respectively, slightly higher compared to the Cr alloy steels. In this case, the contact stress is almost equal to the value of GFRP/Cr alloy steel friction pairs $\sigma_{c(\max)}=4.44\;\mathrm{MPa}$.

The final set of materials examined consisted of a glass fiber composite disc and a technical ceramic ball composed of Al_2_O_3_. Alumina, or aluminum oxide, is a highly durable and wear-resistant material that can cause notable wear to GFRP due to its high hardness and abrasive properties. For this friction pair the contact pressure value is the highest $\sigma_{c(\max)}=27.152\;\mathrm{MPa}$.

The time-dependent behavior of the coefficient of friction was observed during the initial stages of the experiment. The coefficient of friction gradually increased and stabilized after approximately 120 min. The values presented in Figure 14 are the average coefficient of friction obtained over the entire experiment, and the error bars correspond to the standard deviation of three measurements, indicating the variability of the test data. Experimental observations indicate that the coefficient of friction (COF) between two surfaces undergoes an initial increase during the accommodation period of the first 1–3 min of testing, as the sliding speed increases. However, with the passage of time, this trend reverses, and the COF gradually decreases.

In this case, it was observed that the friction coefficient exhibits an inverse relationship with the sliding speed, i.e., as the sliding speed decreases, the friction coefficient tends to increase, namely from 0.415–0.46 for a 0.10 m s^−1^ sliding speed, 0.315–0.48 for a 0.25 m s^−1^ sliding speed, and in the range 0.24–0.47 for a 0.36 m s^−1^, respectively.

Figure 14 shows that as the sliding distance increases, the coefficient of friction with respect to sliding speed decreases and is stabilizing. From the point of view of the local temperature, it decreases significantly with increasing speed, being able to observe that at a speed of 0.1 m s^−1^, the maximum value it reaches is around 68 °C. At a speed of 0.36 m s^−1^, it only reaches 38 °C due to the formation of a thin film of oxide, which acts as a thermal barrier, reducing the heat transfer between two materials and lowering the local temperature. The relationship between sliding speed and local temperature in dry friction is so complex and depends on many factors, including the materials involved, the surface roughness, and the environmental conditions.

### 4.2. Wear Pattern and 3D Optical Images of Worn-Out Ball and GFRP Specimens

When a glass-fiber-reinforced composite interacts mechanically with PTFE, the wear rate of the composite can vary depending on several factors such as the composition and the specific testing conditions. Glass-fiber-reinforced composite materials have high strength and stiffness characteristics, while PTFE has low friction and good chemical resistance. Several studies have been conducted to analyze the wear behaviors of glass-fiber-reinforced composites against PTFE.

Our testing shows that the wear rate of glass-fiber-reinforced composites against PTFE is lower compared to that of other polymer composites. The lower wear rate of the composite against PTFE is attributed to the fact that PTFE has a low coefficient of friction which helps in reducing the contact stress and hence the wear rate. On the other hand, due to the extremely ductile nature of PTFE, it shears off easily and possesses a high wear rate.

Experimental testing has reported that GFRP composites can exhibit higher wear resistance than PTFE in certain conditions, like dry sliding and low-load applications.

It was found that for the GFRP disc, the wear is insignificant; it could not be highlighted either by weighing or by profilometry for any of the sliding speed values. Some very fine traces of wear can be highlighted under Nano Focus—Optical 3D Microscopy on the surface of the disc (Figure 15a,b). While the PTFE ball (Figure 15c) suffered severe wear, obtaining a wear rate of 28.46 × 10^−5^ mm^3^/Nm, 35 × 10^−5^ mm^3^/Nm, and 38.92 × 10^−5^ mm^3^/N m for the sliding speeds of 0.1, 0.25, and 0.36 m s^−1^, respectively. The volume lost through wear is between 4–18 mm^3^.

For the GFRP/Torlon pair (Figure 16), the wear rate obtained for the GFRP disc for the three sliding speed values is around (0.61–6167) × 10^−6^ mm^3^/Nm. For the Torlon ball, the wear rate in the same conditions is 0.583 × 10^−5^ mm^3^/Nm, 0.543 × 10^−5^ mm^3^/Nm, and 0.486 × 10^−5^ mm^3^/Nm. The wear marks on the ball will appear. The wear mark no longer appears so uniform that we can define it as a circle, it has a more uneven shape close to an elliptical mark, in some places as small pits or scratches, and the surface may become dulled or roughened. The volume lost through wear is between 2.24–12.43 mm^3^.

Since resins contain pores, the pores present at the interface determine the amount of mass loss during physical transfer to another material. Mass loss, inherent in all resin materials, is a complex phenomenon. Apart from physical properties, the size, shape, and number of pores present at the contact surface can inevitably affect the mass loss properties of resin materials.

The wear pattern and 3D optical images of worn-out 52100 Chrome alloy steel and GFRP specimens are presented in Figure 17. The wear loss of the disc is observed to be in the range of 0.001–0.003 g with a decrease in sliding speed. In the case of solid metal, the wear loss depends on the density and hardness of the material. Based on the experimental results obtained by profilometry, the wear rate of the composite disc range for the three decreased sliding speed values is around 4.442 × 10^−5^ mm^3^/Nm, 6.12 × 10^−5^ mm^3^/Nm, and 11.456 × 10^−5^ mm^3^/Nm, respectively, and increases with sliding speeds decrease. The wear marks of the Chrome alloy steel ball for decreasing sliding speeds are presented in Figure 18. For the ball, the wear mark increases with increasing speed in the range of 4.4 × 10^−5^ mm^3^/Nm–5.897 × 10^−5^ mm^3^/Nm.

For the 440 Stainless steel against GFRP pair, the disc wear pattern and 3D optical images of worn-out surfaces are presented in the figures below—Figure 19 and Figure 20. The wear loss of the GRFP disc against the chrome alloyed steel ball is observed to be slightly lower than the GRFP/440 stainless steel pair. The highest and lowest mass losses observed for the GFRP/stainless steel are 0.007 and 0.002 g at the 0.36 m s^−1^ to 0.10 m s^−1^ speed values, respectively. The minimum and maximum mass loss values are very small, but in the case of GFRP/440 steel, it has doubled for all three sliding speed values. The wear rates of the GFRP composite disc are 7.614 × 10^−5^ mm^3^/N m, 10.451 × 10^−5^ mm^3^/N m, and 11.3965 × 10^−5^ mm^3^/N m, respectively. The wear pattern observed on the ball surface is no longer circular in shape and exhibits an irregular elliptical shape, characterized by the presence of small pits and scratches. Furthermore, the surface texture shows signs of dullness and roughening. The wear characteristics of 440 stainless steel are dependent on the microstructure and heat treatment of the material. For the ball, the wear mark increases with increasing speed in the range of 4.2 × 10^−5^ mm^3^/N m–5.62 × 10^−5^ mm^3^/N m, values relatively close to the previous case.

In the case of the GFRP/Al_2_O_3_ friction coupling, the wear condition of the disc is shown in Figure 21 and Figure 22. The wear loss of the disc obtained by weighing on the microbalance is in the range of 0.019–0.010 g with the decrease in sliding speed. Under these conditions, for the wear intensity K, values were between 25.723 × 10^−5^ mm^3^/N m, 22.484 × 10^−5^ mm^3^/N m, and 19.65 × 10^−5^ mm^3^/N m, respectively.

Al_2_O_3_ is known for its high hardness, chemical stability, and wear resistance, which typically results in a low wear rate. On the other hand, GFRP has lower wear resistance than Al_2_O_3_ due to its lower hardness and lower resistance to abrasive wear. The wear rate (K) of Al_2_O_3_ as expected is lower compared to GFRP in a pin-on-disc wear test due to its higher wear resistance.

On the alumina balls under the same test conditions, no sign of wear was seen, it was then analyzed under a very high-power microscope and very small scratches and rather deposits from the composite material were highlighted, as seen in Figure 22. During the sliding process, the contact between the Al_2_O_3_ ball and the GFRP disk generates heat and friction, which can cause the resin in the composite material to soften and transfer onto the ball surface. The extent of resin transfer depends on the quality of the resin, the roughness and hardness of the ball surface, the contact pressure and speed, and the duration of the sliding process.

Figure 23, Figure 24 and Figure 25 show the comparative values of the friction coefficients for the five ball/disc friction pairs depending on the sliding speeds v_1_ = 0.36 m s^−1^, v_2_ = 0.25 m s^−1^, and v_3_ = 0.10 m s^−1^, respectively.

Results showed that the coefficient of friction for all pairs decreased with an increase in sliding speed.

The comparative values of wear rate K for the GFRP disk in the five cases of friction pairs are depicted in Figure 26, with respect to the sliding speed. And in Figure 27 is presented the comparative values of wear rate K for balls for peripheral speeds at test time (120 min). The wear rate K between GFRP and PTFE is subject to various factors, including sliding speed. However, determining the wear rate of GFRP against PTFE experimentally at sliding speeds is challenging due to several reasons. These reasons include the low coefficient of friction of PTFE, which may not provide sufficient energy dissipation to cause significant wear of the GFRP material, particularly at higher sliding speeds, and the self-lubricating properties of PTFE, which can reduce friction and wear between the two surfaces in contact. These properties make it difficult to quantify the wear rate of GFRP against PTFE, which may be negligible. Furthermore, GFRP and PTFE have significantly different mechanical properties, which further complicates the determination of the wear rate. Similar challenges may arise when determining the wear rate of the GFRP disc/Torlon ball material pair.

The wear rate K, between GFRP and chrome alloy steel and stainless steel, can be affected by various factors, including sliding speed. In general, as sliding speed increases, the wear rate between GFRP and chrome alloy steel and stainless decrease.

One possible explanation for this behavior is the effect of sliding speed on the formation of a transfer layer. At low sliding speeds, the contact between the pin and disc can lead to the accumulation of debris, which can promote abrasive wear and increase the wear rate. However, with how much the sliding speed increases, the debris was partly dispersed from the contact area and partly transferred to the surface of the counter material, leading to the formation of a transfer layer that reduces the wear rate.

The experiments demonstrate that the wear rate K between GFRP and Al_2_O_3_ increases with sliding speed, contrary to the expectation of a decrease. This phenomenon can be attributed to the fact that at higher sliding speeds, the contact pressure and temperature at the interface between the two materials increase. This rise in the contact pressure and temperature can cause a corresponding increase in the rate of material removal, leading to an increase in the wear rate.

At higher sliding speeds, the increased contact pressure and frictional forces can cause more severe wear and abrasion, which can lead to a higher wear rate. In addition, the high temperature generated at the contact interface can cause thermal softening and degradation of the polymer matrix in GFRP, leading to accelerated wear.

From the analysis of the wear behavior of the five balls of different materials, we conclude that with the increase of the sliding speed, the contact pressure and temperature between the ball and the surface of the GFRP disk increases. This can lead to more severe wear and deformation of the ball surface. At higher speeds, the wear mechanism shifts from adhesive wear to abrasive wear, where the harder and rougher particles on the disc surface cause more damage to the ball surface.

## 5. Conclusions

Experimental results conducted in this work allowed the following conclusions:

Friction-related outcomes:

-Initially, the friction coefficient increases with operating time but eventually reaches a stable value that remains relatively constant.-With an increase in sliding distance, the friction coefficient values change marginally and remain higher for dry conditions.-As the sliding velocity increases, the friction coefficient decreases; however, the rate of decrease slows down at higher velocities.-The time required to reach a stable value of the friction coefficient is independent of the bearing pressure.-Under dry testing conditions, the coefficient of friction ranges between 0.18 to 0.58 for different friction pairs, working conditions, and sliding distances.

Wear-related outcomes:

-At the start of the operation, the wear rate increases rapidly with operating time, and this corresponds to a sliding distance of 2.593 km. Under dry testing conditions wear pattern increases considerably with sliding distance.-The wear rate increases with an increase in velocity, although the rate of increase decreases as the velocity further increases. For dry testing conditions, wear value ranges from 0.009 mm^3^ up to 13–13.5 mm^3^ for GFRP disc in working conditions. For balls wear values range between 0.001 mm^3^ up to 39–39.5 mm^3^.-Overall, while the wear rate K between GFRP and PTFE is affected by sliding speed, so the wear rate of GFRP against PTFE is undefined or difficult to determine experimentally at high sliding speeds due to the low coefficient of friction, self-lubricating properties, and potential for adhesive wear. It is a similar situation for Torlon.-The wear rate K, between GFRP and chrome alloy steel, and stainless steel decreases as sliding speed increases. The contact between the pin and disc can lead to the accumulation of debris, which can promote abrasive wear and increase the wear rate.-For the GFRP and Al_2_O_3_, the wear rate K increases with sliding speed rather than decreasing. Because the contact pressure and the temperature at the interface between the materials also increase, it can result in an increase in the rate of material removal and a corresponding increase in wear rate.-From the analysis of the wear behavior of the balls, it was seen that as the sliding speed increases, more severe wear and deformation of the ball surface occurs. the wear mechanism shifts from adhesive wear to abrasive wear, where the harder and rougher particles on the disc surface cause more damage to the ball surface.-It is important to note that the relationship between sliding speed and wear rate can depend on various factors, including the specific test conditions, the materials being tested, and the nature of the wear mechanisms involved. Therefore, the effect of sliding speed on wear rate should be evaluated on a case-by-case basis.-To accurately assess the wear behavior of a ball on a disc pin, the wear marks were carefully analyzed with optical profilometry, thus providing valuable information on wear mechanisms that will help optimize system design and performance.-The selection of the material will depend on the specific application requirements and operating conditions.

The composite material presented in this work has not yet been studied from the point of view of tribomechanical properties. This is a starting point for future research, in which new constituents will be introduced into the combination of studied materials. The optimization of the manufacturing parameters and tribomechanical testing is being pursued to improve the material in terms of wear and friction.

## Figures and Tables

**Figure 1 polymers-15-02733-f001:**
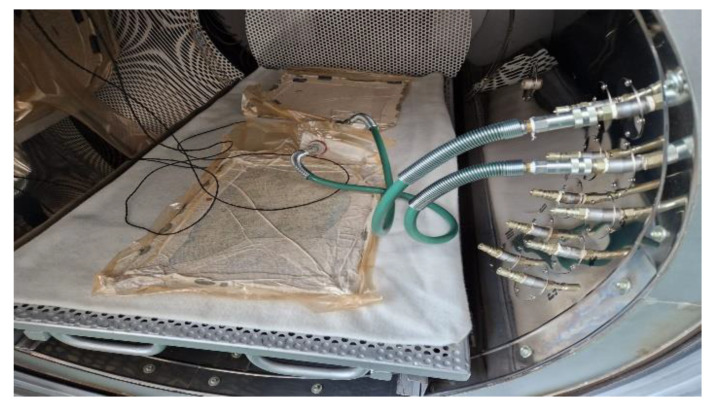
Autoclave curing procedure of GFRP samples.

**Figure 2 polymers-15-02733-f002:**
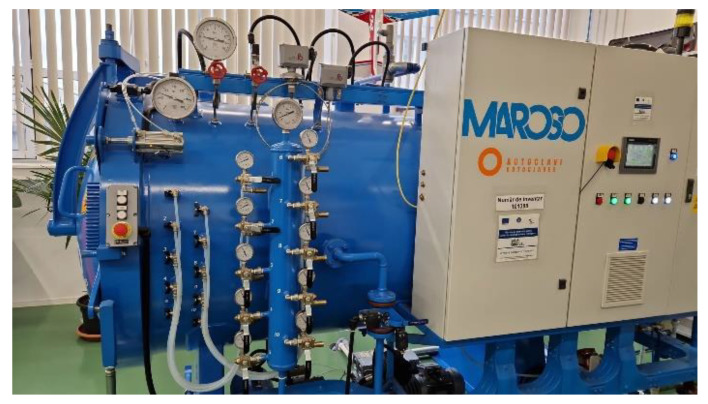
Maroso autoclave.

**Figure 3 polymers-15-02733-f003:**
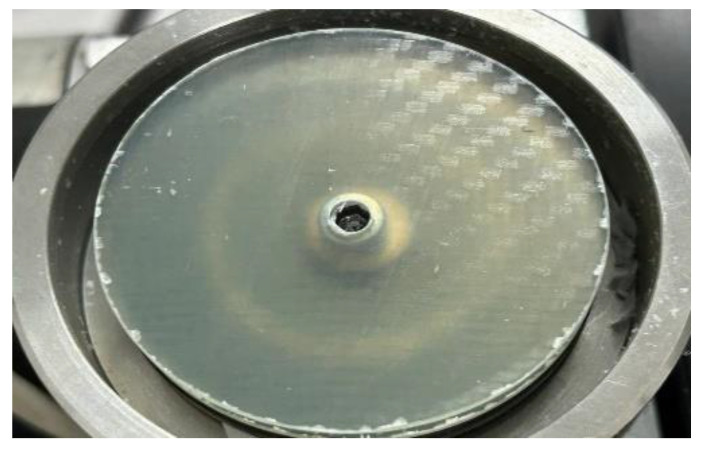
Glass fiber composite specimen (disc).

**Figure 4 polymers-15-02733-f004:**
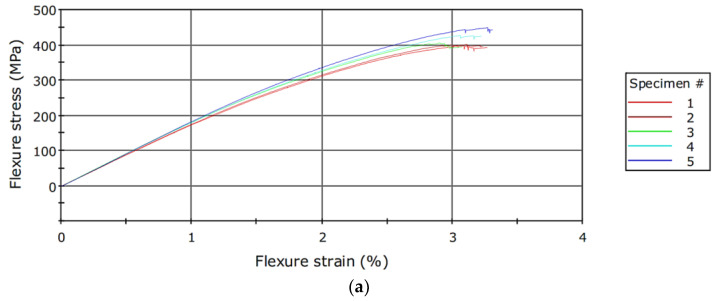
(**a**). Stress–strain curves for flexural tests; (**b**) Stress–strain curves for tensile tests.

**Figure 5 polymers-15-02733-f005:**
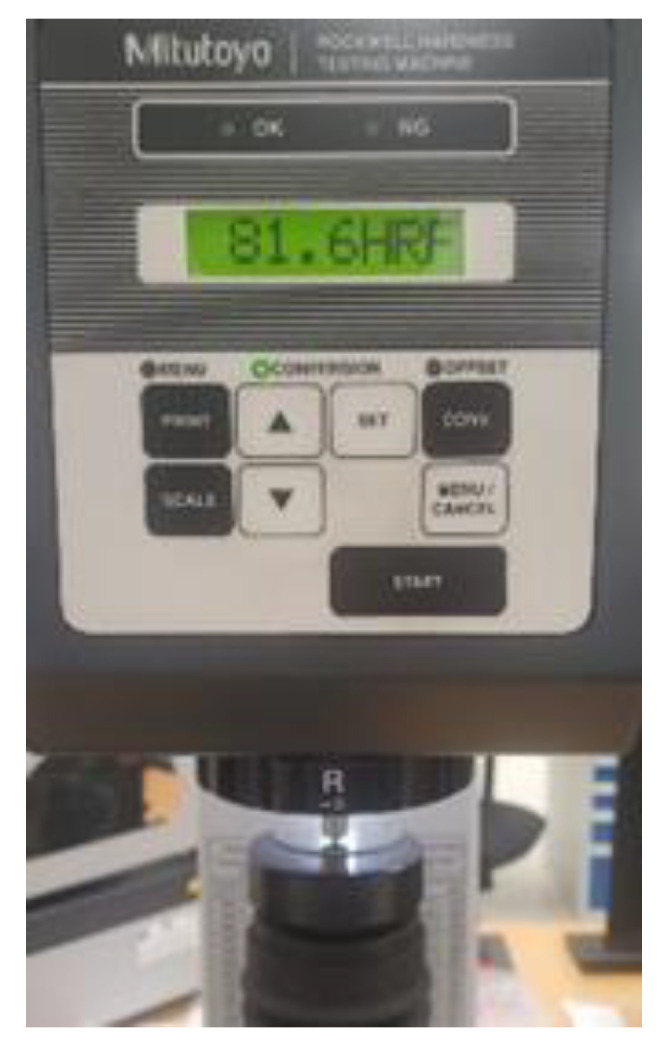
Hardness measurement.

**Figure 6 polymers-15-02733-f006:**
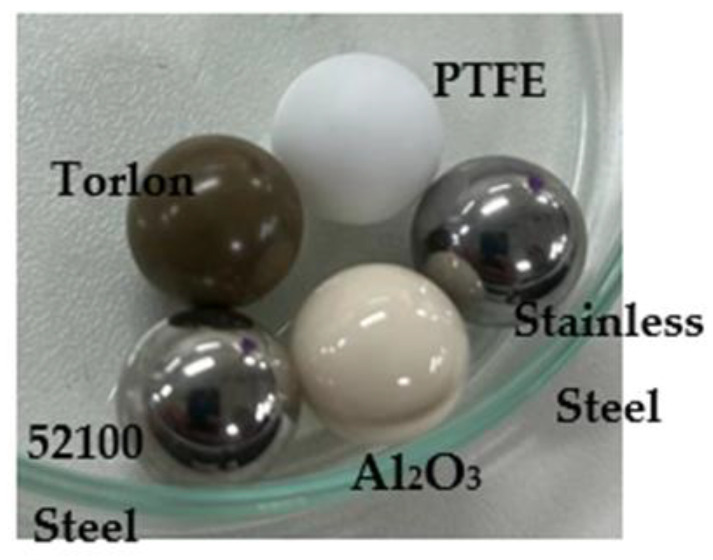
Bearing balls (12.7 mm diameter).

**Figure 7 polymers-15-02733-f007:**
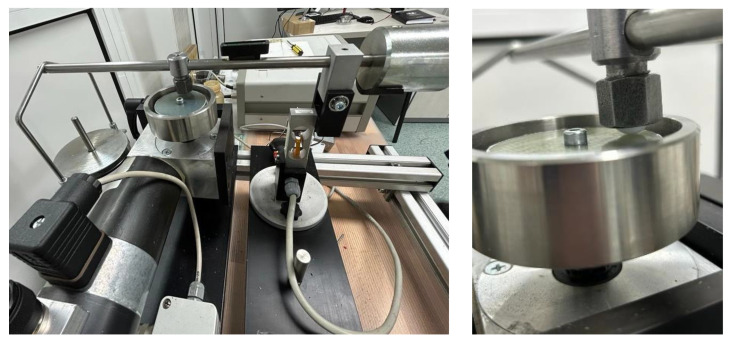
A modified pin-on-disc tribometer.

**Figure 8 polymers-15-02733-f008:**
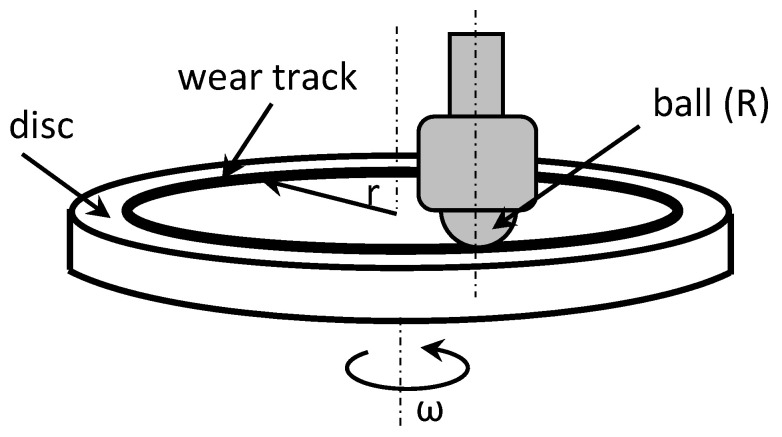
Pin-on-disc tribometer scheme (wear track model).

**Figure 9 polymers-15-02733-f009:**
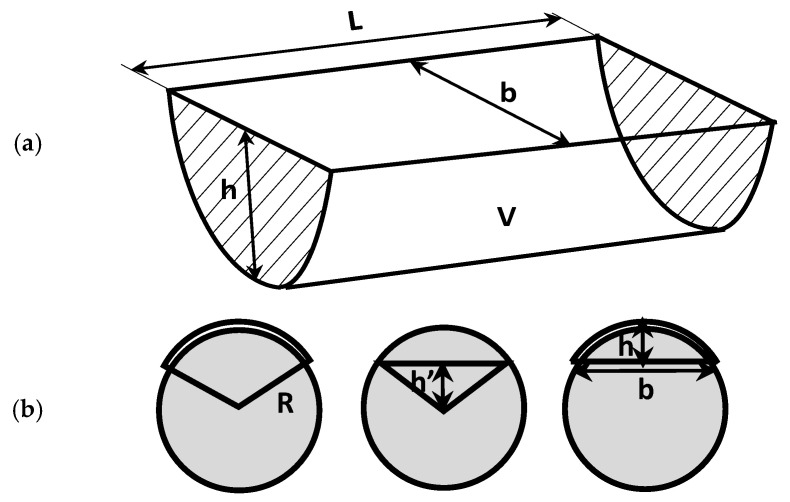
Model for calculating wear track volume: (**a**) on the disc; (**b**) on the ball.

**Figure 10 polymers-15-02733-f010:**
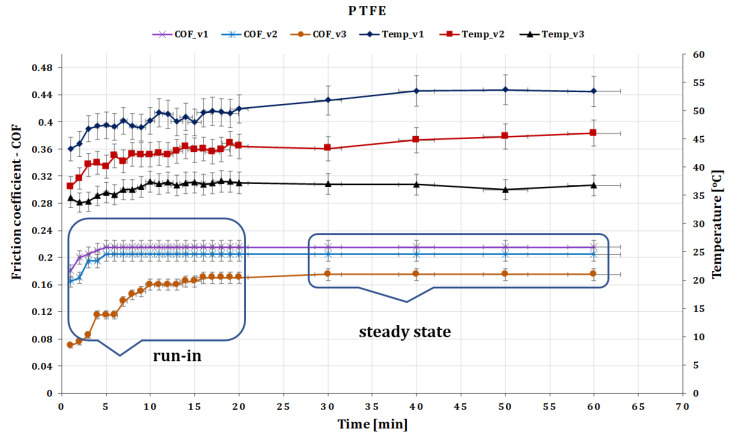
The variation of friction coefficient of GFRP composite against PTFE versus peripheral speed at test time (60 min).

**Figure 11 polymers-15-02733-f011:**
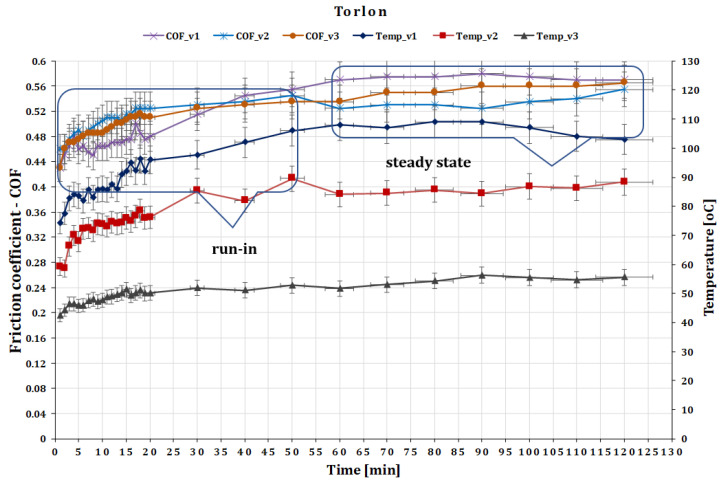
The variation of friction coefficient of GFRP composite against Torlon versus peripheral speed at test time (120 min).

**Figure 12 polymers-15-02733-f012:**
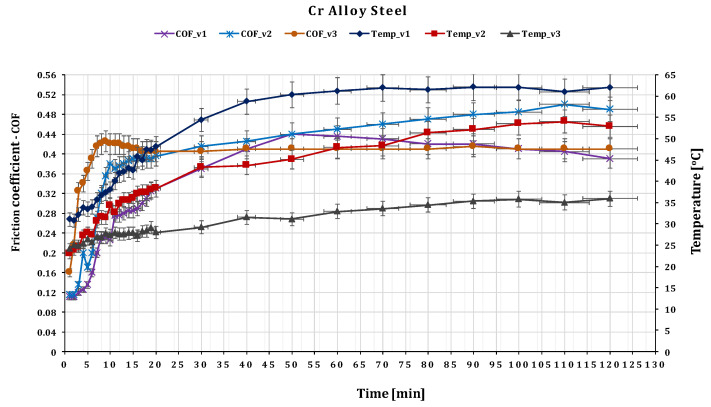
The variation of friction coefficient of GFRP composite against 52,100 Chrome Alloy Steel versus peripheral speed at test time (120 min).

**Figure 13 polymers-15-02733-f013:**
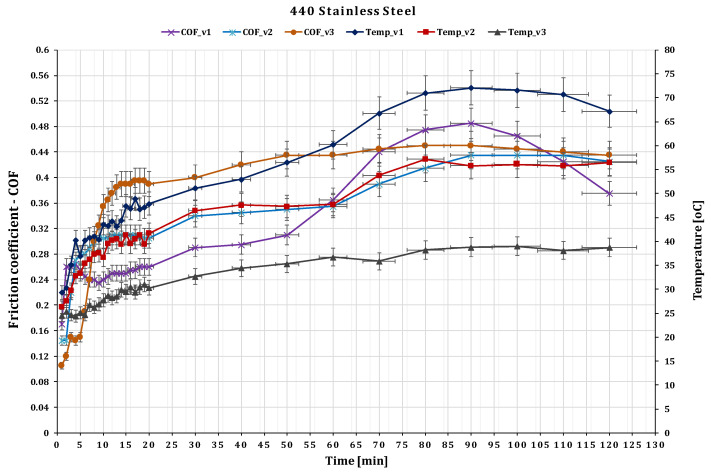
The variation of friction coefficient of GFRP composite against 440 Stainless Steel versus peripheral speed at test time (120 min).

**Figure 14 polymers-15-02733-f014:**
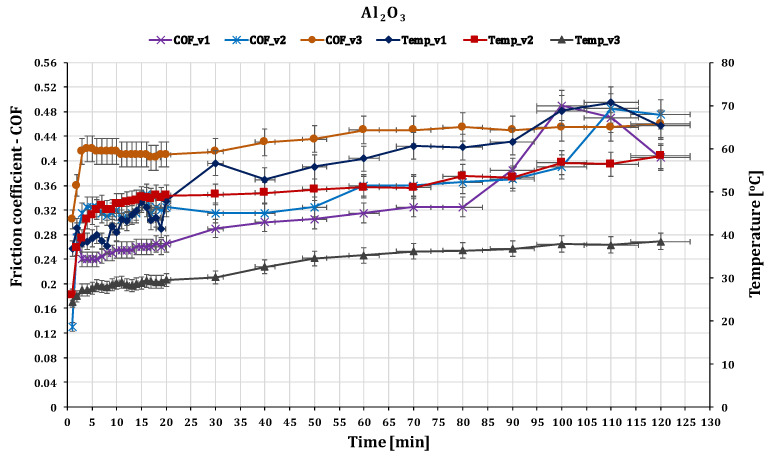
The variation of friction coefficient of GFRP composite against Al_2_O_3_ versus peripheral speed at test time (120 min).

**Figure 15 polymers-15-02733-f015:**
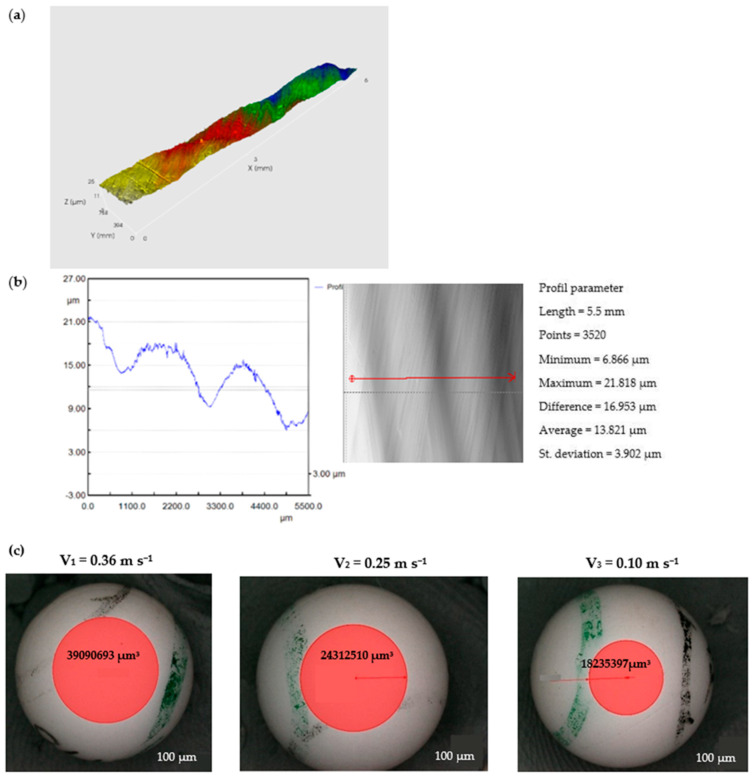
(**a**) The worn surfaces morphologies of GFRP after friction experiment dry lubricated under F_n_ = 20 N, duration 120 min, (**b**) Wear surface profile curve for disc (**c**) Wear marks of PTFE ball for different sliding speeds.

**Figure 16 polymers-15-02733-f016:**
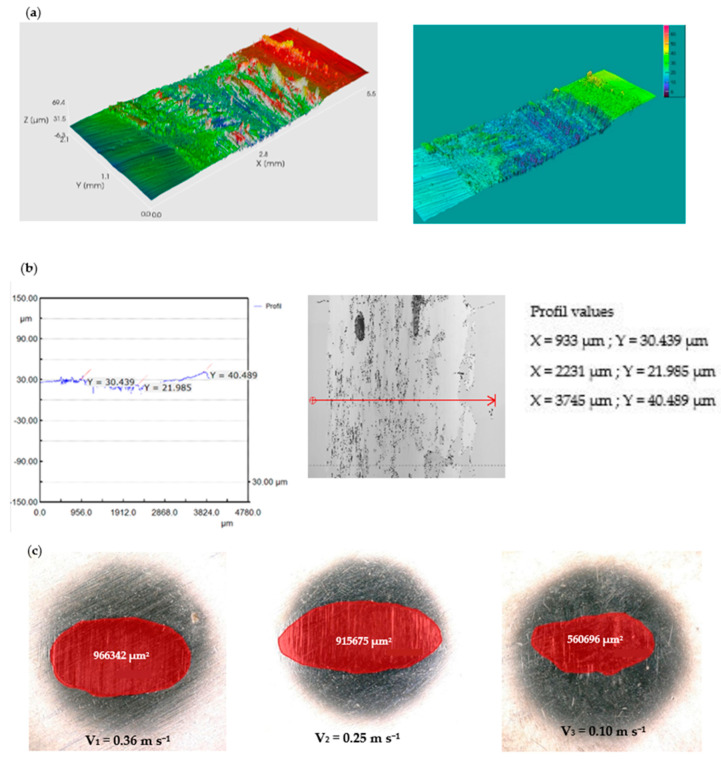
(**a**) The worn surface morphologies of GFRP after friction experiment dry lubricated under Fn = 20 N, duration 120 min; (**b**) Wear surface profile curve for disc; (**c**) Wear marks of Torlon ball for different sliding speeds.

**Figure 17 polymers-15-02733-f017:**
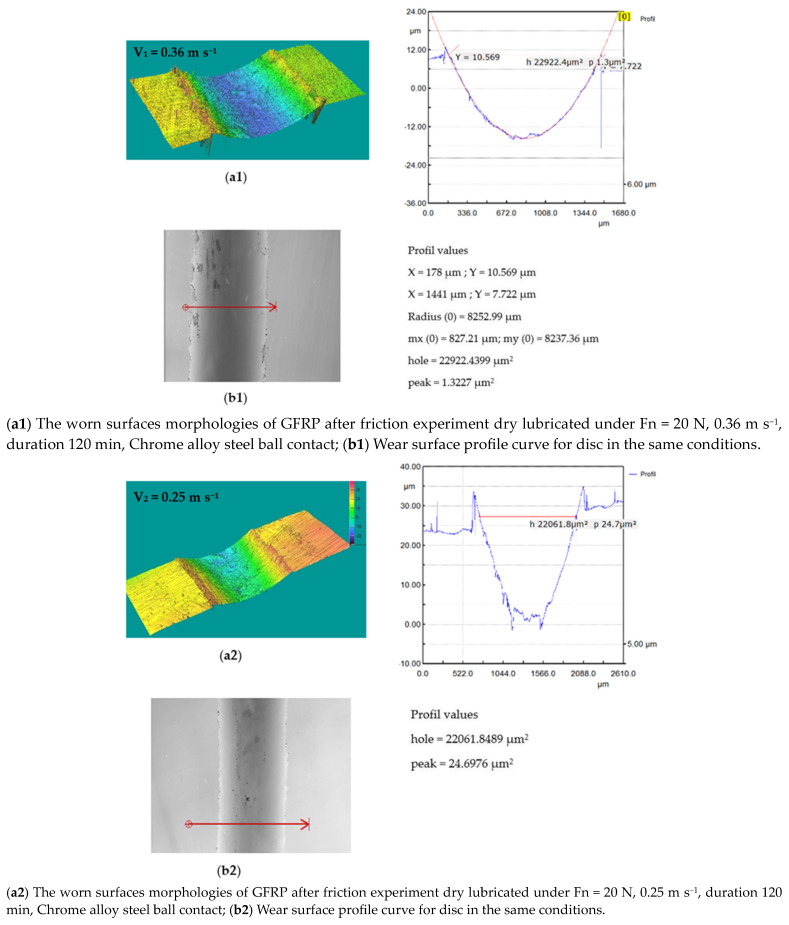
(**a1**–**a3**) The worn surfaces morphologies of GFRP after friction experiment dry lubricated under F_n_ = 20 N, three sliding speeds, duration 120 min, Chrome alloy steel ball contact; (**b1**–**b3**) Wear surface profile curve for disc in the same conditions.

**Figure 18 polymers-15-02733-f018:**
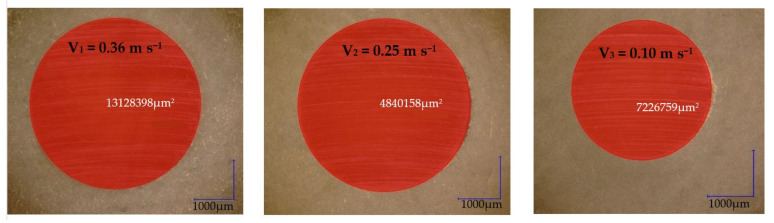
Wear marks of Chrome alloy steel ball for different sliding speeds.

**Figure 19 polymers-15-02733-f019:**
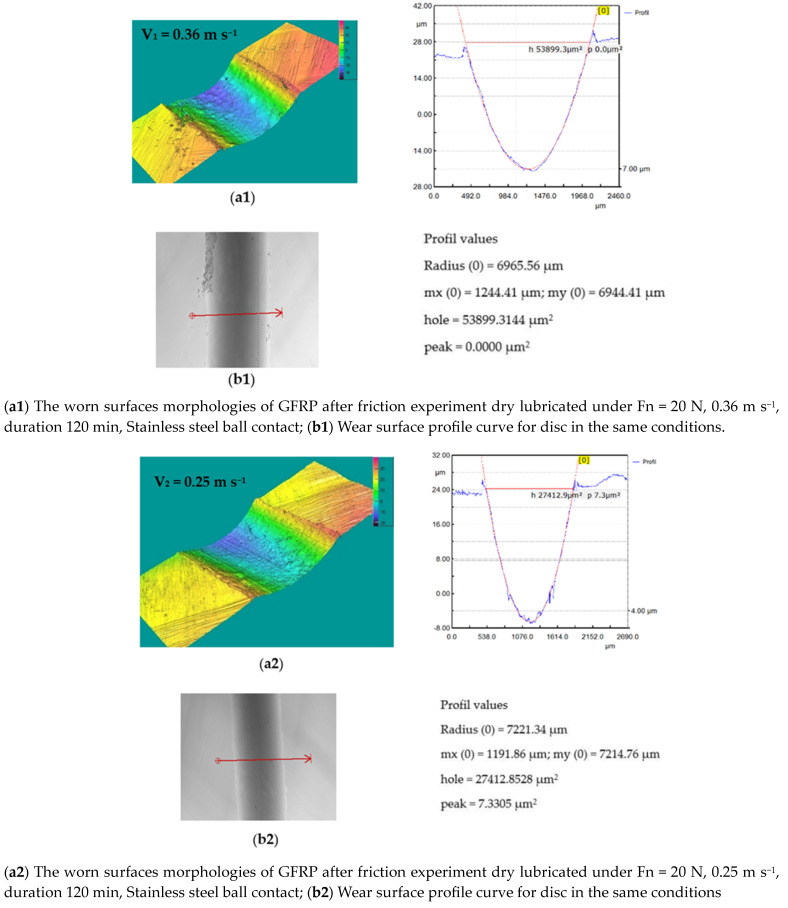
(**a1**–**a3**) The worn surfaces morphologies of GFRP after friction experiment dry lubricated under F_n_ = 20 N, three sliding speeds, duration 120 min, Stainless steel ball contact; (**b1**–**b3**) Wear surface profile curve for disc in the same conditions.

**Figure 20 polymers-15-02733-f020:**
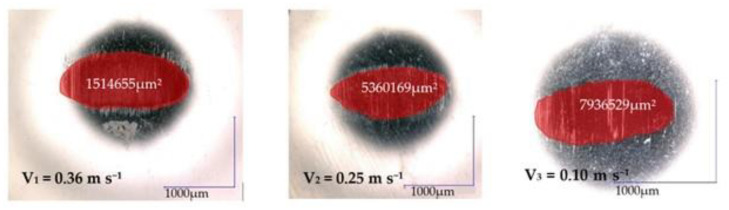
Wear marks of 440 stainless steel balls for different sliding speeds.

**Figure 21 polymers-15-02733-f021:**
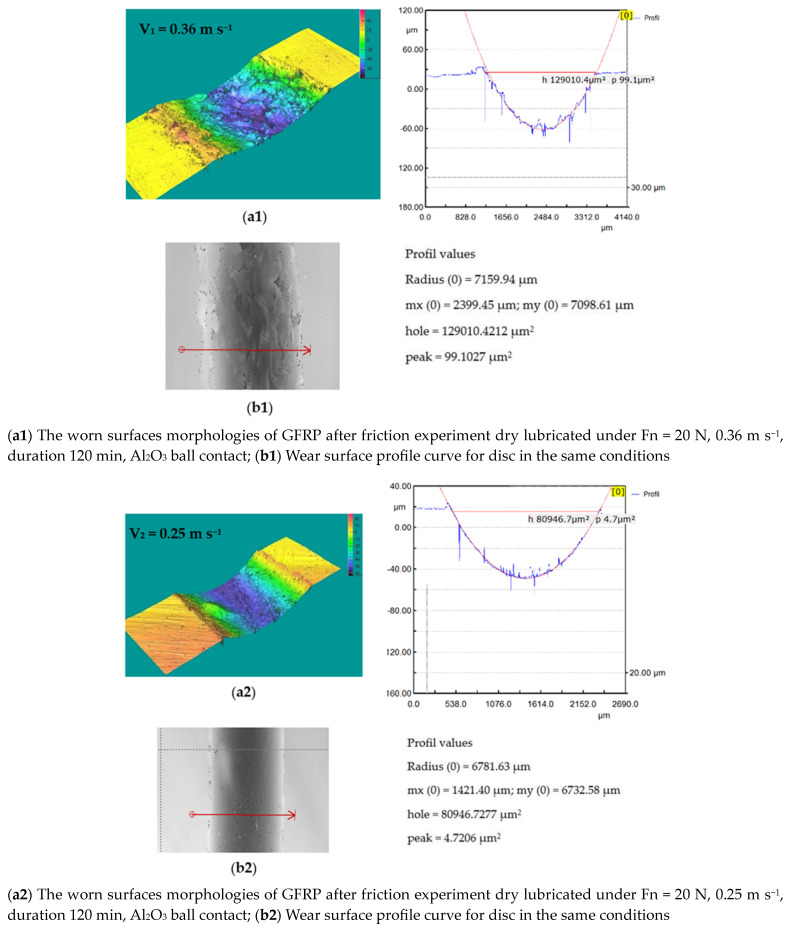
(**a1**–**a3**) The worn surfaces morphologies of GFRP after friction experiment dry lubricated under F_n_ = 20 N, three sliding speeds, duration 120 min, Al_2_O_3_ ball contact; (**b1**–**b3**) Wear surface profile curve for disc in the same conditions.

**Figure 22 polymers-15-02733-f022:**
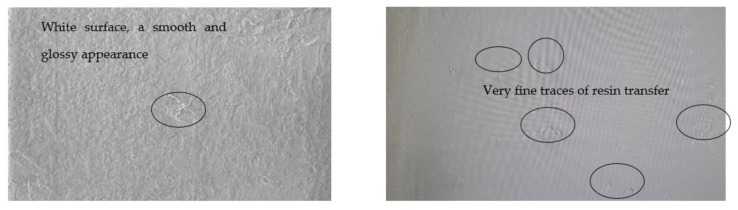
Wear marks of Al_2_O_3_ ball.

**Figure 23 polymers-15-02733-f023:**
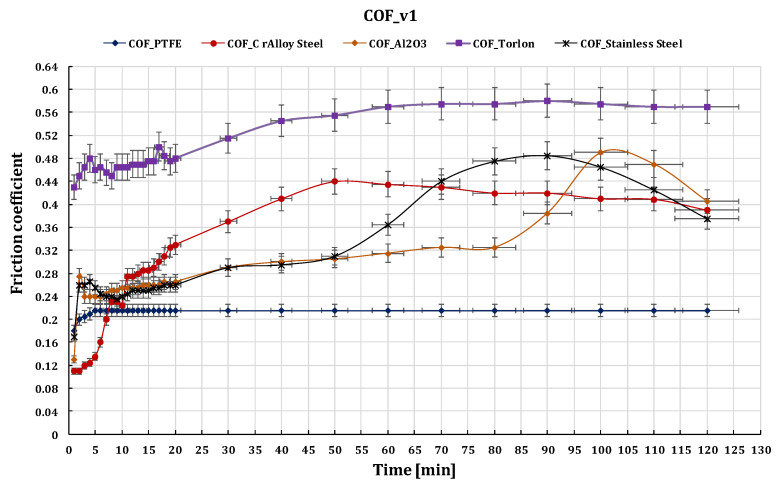
The variation of friction coefficient of GFRP composite against five types of balls versus 0.36 m s^−1^ peripheral speed at test time (120 min).

**Figure 24 polymers-15-02733-f024:**
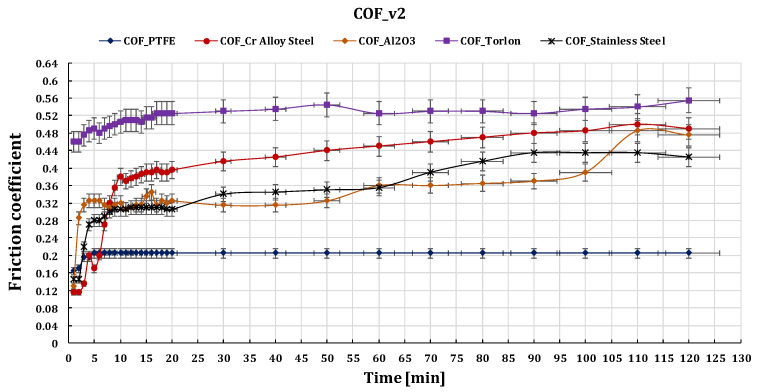
The variation of friction coefficient of GFRP composite against five types of balls versus 0.25 m s^−1^ peripheral speed at test time (120 min).

**Figure 25 polymers-15-02733-f025:**
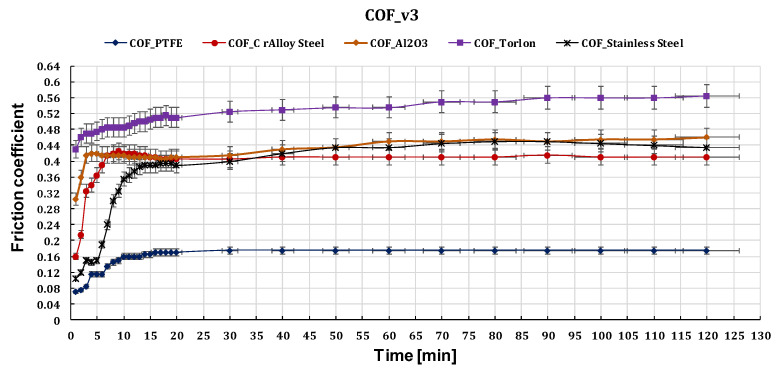
The variation of friction coefficient of GFRP composite against five types of balls versus 0.10 m s^−1^ peripheral speed at test time (120 min).

**Figure 26 polymers-15-02733-f026:**
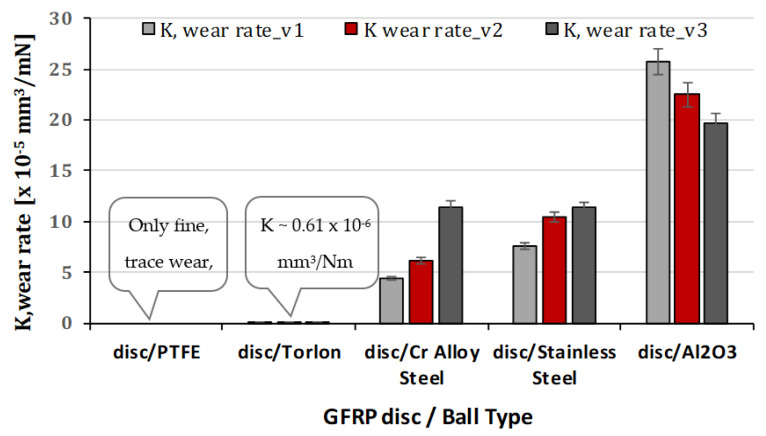
Wear rate K for GFRP discs for peripheral speeds at test time (120 min).

**Figure 27 polymers-15-02733-f027:**
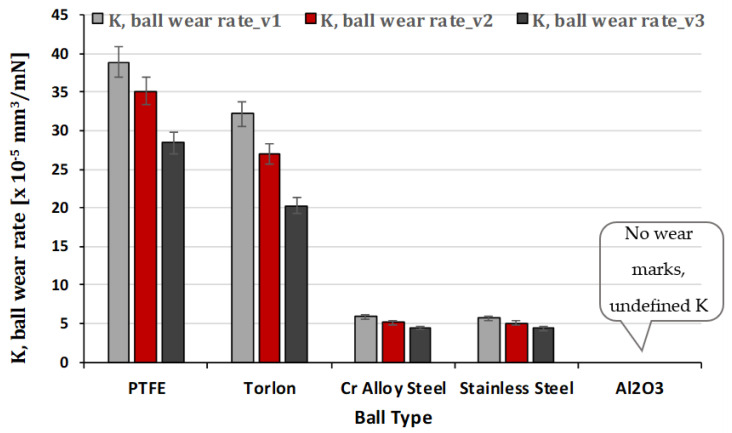
Wear rate K for balls for peripheral speeds at test time (120 min).

**Table 1 polymers-15-02733-t001:** Flexural proprieties of glass fiber-reinforced polymer composites.

Specimen Type	Flexure Stress (MPa)	Flexure Strain (%)	E-Modulus (MPa)
GFRP 68.5%	415.56 (21.56)	3.087 (0.129)	18,218.1 (391.6)

**Table 2 polymers-15-02733-t002:** Tensile proprieties.

Specimen Type	Tensile Strength (MPa)	Tensile Strain at Tensile Strength (mm/mm)	E-Modulus (MPa)
GFRP 68.5%	480.1 (25.92)	2.941 (0.143)	22,181.7 (253.2)

**Table 3 polymers-15-02733-t003:** Chemical and mechanical properties of ball material.

Ball Type (12.7 mm)	Chemical Composition [%]	Mechanical Properties					
		Hardness HRC Scale	Compressive Strength (MPa)	Yield Strength (MPa)	Young’s Modulus (GPa)	Poisson’s Ratio	Roughness R_a_ (µm)
Polytetrafluoroethylene PTFE (Teflon) ρ = 2.2 g/cm^3^	a strong, tough, waxy, nonflammable synthetic resin	50 D	24.5	13.8–15.2	1.45	0.46	0.64–0.78
Torlon 4200 ρ = 1.42 g/cm^3^	Unreinforced, unpigmented grade of polyamide-imide (PAI) resin	80 E	221	150	4.2	0.45	0.46–0.53
52100 Chrome Alloy Steel ρ = 7.81 g/cm^3^	Fe: 96.5–97.3 C: 0.98–1.1 Si: 0.15–0.35 Cr: 1.4–1.6 Mn: 0.25–0.45 P and Si	54–58	2100–2200	2000	200	0.3	0.282–0.30
440 Stainless Steel ρ = 7.7 g/cm^3^	Fe: 96.5–97.3 C: 0.95–1.12 Si: 1 Cr: 16–18 Mn: 1 Mo, P and Si	58–65	2100–2200	1900	200	0.275	0.307–0.33
Alumina Oxide Ceramic Al_2_O_3_ ρ = 3.8 g/cm^3^	Al_2_O_3_: 98.6 SiO_2_: 0.18–0.2 CaO: 0.2 Fe_2_O_3_: 0.02 TiO_2_: 0.02	85	2400		380	0.25	0.22–0.28

**Table 4 polymers-15-02733-t004:** Working parameters for experiments.

Parameters	Operating Conditions
Normal load	20 N
Sliding velocity	0.1, 0.25, 0.36 m s^−1^
Rotating speed	Max 215 (±3) rpm
Relative humidity	45 (±5)%
Starting temperature (RT)	22 (± 2) °C
Duration of rubbing	120 min
Surface conditions	Dry lubrication
Disc/ball material	Glass fiber reinforced polymer (GFRP) composite/Chrome Alloy Steel, Stainless steel, alumina, Teflon, Torlon
Average surface roughness R_a disc_	0.38 µm

## Data Availability

All the data are available with the authors and can be provided on request. Correspondence and requests for materials should be addressed to C.B. and M.C.

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
