# Peer review of "Tribo-Mechanical Investigation of Glass Fiber Reinforced Polymer Composites under Dry Conditions"

_polymers, 2023, doi:10.3390/polym15122733_

Round 1

Reviewer 1 Report

The paper entitled ‘’ Tribo-mechanical investigation of glass fiber reinforced polymer composites under dry conditions’’ deals with study of mechanical properties of glass reinforced polymer composites and the sliding friction properties of GFRP affect the material’s performance. The work is very interesting however there are some points that need to be addressed in order to be publishable, according to the reviewer’s opinion.

1.      In the introduction part the contribution and the innovation of the work should be stated in a clearer way. The authors present previews work (which is necessary) however they should declare how their work will go one step further in this direction.  In addition, the references are poor, authors should definitely enriched them.

2.      The fabrication procedure is presented in detailed, however I would like to ask why the reinforcement in GFRP should be 68.5%. Is there any preliminary study which suggests reinforcement.

3.      The variation of sliding speed effect on friction and wear is very interesting section. In page 10  the authors refer  ‘the coefficient of friction of PTFE decreases with increasing speed. This behavior may be attributed to the formation of a self lubricating film on the surface of PTFE when in contact with other materials. The membrane is formed by shear-induced orientation of PTFE polymer chains, reducing adhesion between contact surfaces, and lowering the coefficient of friction’’.   Is there any other way to ensure that this beahvior is assigned to the film created on the surface of PTFE?

Maybe AFM or some other microscopy coupling technique?

4.      The conclusion should be rearrange and express  in a more sufficient way the goal and the innovation of the results.

The quality of Engish language is quite good and requires only some moderate editing. 

Reviewer 2 Report

Based on the results presented in the study, the following technical comments are to be addressed:

1-      The number of references in the text does not match the ones listed in the Section of ‘References’;

2-      There are several types of glass fibers. The one used in this study should be specified in the ‘Abstract’;

3-      The Keywords should represent the method and results of the study. Please add additional words to the ‘Keywords’;

4-      Please clearly identify the objective/s of the current study;

5-      Please remove the paragraph in lines 100-105;

6-      The authors carried out the various tests using a pre-prepared specimen. What about having out-of-shelf specimens in order to produce reliable results? Please discuss;

7-      The curves shown in Fig. 5 are poor and not clear;

8-      The number of references listed in this study is very limited (18 references) although the study is conducted in a well established area;

9-      In line 230, the authors mentioned that ‘m’ is the weight lost. Please modify the definition to be the ‘mass’;

10-  In line 231, please modify ‘ρ’ to match the one provided in eqn 1;

11-  Please use different numbers to indicate equations 3;

12-  In line 267, the authors specified the value of σcmax to be 0.7363 MPa. What is the basis of selection?

13-  In lines 268-269, the authors mentioned that the coefficient of friction (COF) decreases ….’ there is no decrease in the COF as can be seen in Fig. 10. The decrease is in Temp-v1 to Temp-v3. Please revise;

14-  In lines 286-287, the authors refer to the literature. Please indicate the specific reference/s;

15-  Please explain the reason of having different values of σcmax;

16-   Generally, the current study investigates the coefficient of friction of GFRP laminate using different balls. Is there previous study/ies conducted in this subject? If yes, please compare its/their results with the ones obtained in this study;

17-  The results presented in Figs. 10-12 for the time vs coefficient of friction are steady, whereas the results presented in Figs 13 and 14 showed continuous increase till failure. Please discuss;

18-   The conclusion provided in lines 493-494 about the coefficient of friction for GFRP laminate when using different balls is misleading and weak. How come the effect due to speed is significant? For instance, the change in PTFE specimen is between 0.22 to 0.20 when the speed changed from 0.36 to 0.25. Please discuss.

No editorial comments

Author Response

Manuscript Polymers-2421227

Dear Ms. Murphy Wang

Assistant Editor

Polymers Editorial Office

Thank you for giving us the opportunity to submit a revised draft of our manuscript entitled Tribo-mechanical investigation of glass fiber reinforced polymer composites under dry conditions to Polymers Journal, special ISSUE: Mechanical Behavior of Polymeric Materials: Recent Trends and Advancements.

We appreciate the time and effort that you and the reviewers have dedicated to providing your valuable feedback on our manuscript. We are grateful to the reviewers for their insightful comments on our manuscript. We have been able to incorporate changes to reflect most of the suggestions provided by the reviewers. We have highlighted the changes within the manuscript.

In the pdf attached is the point-to-point response to the reviewers’ comments and concerns.

Reviewer 3 Report

This paper tested the sliding friction properties of GFRP against different materials and parameters. This work is interesting and has been well presented. The following questions should be revised before future publication.

1. Abstract:the novel experimental findings can be added in the abstract.

2. Introduction: the writing of the current research review is fine. However, the research purpose is not well refined from the current literature review. Improve the novelty of your experiments. Besides, the Fabrication Procedure of GFRP sample is similar to CFRP, a paper (https://doi.org/10.1016/j.jmrt.2022.12.054) also mentioned the topic of “In the next step the composite material was covered with a perforated film and an absorbent felt, ….”. This paper can be added here to support your method.

3. In Fig. 9, to calculate the wear track volume, the ball and disc groove should be pointed out in the figure. In formular (2): the triangle's height of h' might be replaced by h. The reference should be given for formular (3).

4. Is the selection of test loads and sliding speeds consistent to the industry practice?

5. Conclusion: some outcomes are not novel enough, which can be obtained from the publications decades ago or even from some professional books. So, improve it by further refining the novel experimental findings.

The quality of the language is ok. In general, the language could be refined carefully with the help of a native English speaker. Some writing need to be corrected, such as in “a sampling length of 0,8 mm an evaluation 203 length of 4 mm”, “and” is missed; “the temperature decreases by 52-53°C to 36-37°C” etc.

Author Response

(The authors gave the same response as above.)

Round 2

Reviewer 1 Report

The authors have sucessfully addressed all the reviewer's comments that is why i suggest the manuscript publishable in this present form.